# Evaluating the cost-effectiveness of testing pregnant women for penicillin allergy

Viengneesee Thao[1], Emily E. Sharpe[2], Ruchita Dholakia[1], Hannah H. Ahn[1], James P. Moriarty[1], Bijan J. Borah[1,3], Margaret C. Gill[4], Regan N. Theiler[5]*

**1** Robert D. and Patricia E. Kern Center for the Science of Health Care Delivery, Mayo Clinic, Rochester, Minnesota, United States of America, **2** Department of Anesthesiology and Perioperative Medicine, Mayo Clinic, Rochester, Minnesota, United States of America, **3** Division of Health Care Delivery Research, Mayo Clinic, Rochester, Minnesota, United States of America, **4** Department of Family Medicine, Mayo Clinic, Rochester, Minnesota, United States of America, **5** Department of Obstetrics and Gynecology, Mayo Clinic, Rochester, Minnesota, United States of America

* Theiler.Regan@mayo.edu

## Abstract

### Introduction

True penicillin allergy is rare and is commonly incorrectly reported. In fact, less than five percent of patients who report a penicillin allergy will have a currently active clinically-significant IgE- or T-cell-mediated hypersensitivity when appropriately tested. Penicillin is the agent of choice for intrapartum antibiotic prophylaxis to reduce the risk of group B streptococcus early-onset disease in the newborn. Inaccurate penicillin allergy status may lead to inappropriate antibiotic use, as most alternative drugs are more expensive and broader spectrum than penicillin. Penicillin allergy testing has been found to be safe in pregnancy and cost-effective in other patient populations.

### Objective

To evaluate the cost-effectiveness of penicillin allergy testing and appropriate antibiotic treatment (test then treat strategy) compared to usual care among pregnant women.

### Methods

We developed a decision tree to evaluate the cost of providing appropriate care via a test then treat strategy for pregnant women who report a penicillin allergy, compared to usual care.

### Results

Using the test then treat strategy the additional cost to ensure appropriate care for all pregnant women who report a penicillin allergy, was $1122.38 per person. Adopting a test then treat strategy increased the number of appropriate antibiotic use from 7,843/10,000 to 10,000/10,000 simulations.

**Data Availability Statement:** Most of the data used in our model is available publicly and we have cited those sources accordingly. The only data not publicly available is the cost of antibiotics data which was pulled from patients admitted to Mayo

Clinic in Rochester, MN. The cost data are stored on safe servers at Mayo Clinic, USA and handled confidentially. If access to this data is needed, application for access may be made to the Economic Evaluation Service at the Kern Center for the Science of Health Care Delivery (ees@mayo.edu). Upon approval, applicants would be able to access these data in the same manner as the authors.

**Funding:** MG and RT received competitive funding from the Mayo Clinic Office of Translation to Practice. The funders had no role in study design, data collection and analysis, decision to publish, or preparation of the manuscript.

**Competing interests:** Dr. Theiler has a know-how agreement and research funding from HeraMed and serves on the Medical Advisory Board for Delfina Care. Dr. Borah is a consultant to Exact Sciences and Boehringer-Ingelheim on unrelated (non-Ob/Gyn) projects. This does not alter our adherence to PLOS ONE policies on sharing data and materials.

## Conclusion

Our results show that a test then treat strategy for pregnant women who report a penicillin allergy is a good-value intervention.

## Introduction

Penicillin allergy is the most commonly reported allergy in pregnant women. The reported rate of unconfirmed penicillin allergy of about 8.0% in pregnant women is lower than commonly reported rates in outpatients using healthcare or other hospitalized populations [1, 2]. In hospitalized patients, penicillin allergy is also the most commonly reported antibiotic allergy, occurring in 10–25% of patients [3–6]. When reported adverse reactions to antibiotics are not explored by prescribing providers, it often results in the use of alternative broader spectrum treatment than is required. This practice contributes to the overuse of broad-spectrum agents which has added to the emergence of multiple drug-resistant microbes. Interestingly, even among patients with a history of antibiotic-associated anaphylaxis, only 20–50% will have a positive skin test when tested within 3 months of the reported anaphylaxis [7]. Ultimately, less than five percent of patients who report a penicillin allergy actually have a clinically significant IgE- mediated or T- lymphocyte mediated hypersensitivity when tested [2].

Penicillin is the agent of choice for intrapartum antibiotic prophylaxis to reduce the risk of Group B streptococcus (GBS) early-onset disease in the newborn because it has a narrow, targeted spectrum of antimicrobial activity and is highly effective. An alternative antibiotic to penicillin includes first-generation cephalosporins such as cefazolin for women with a low-risk penicillin allergy. In women with high-risk penicillin allergy or a positive allergy test, clindamycin is recommended; however, GBS isolates can be resistant to clindamycin and therefore susceptibility testing is necessary. When a GBS isolate is not susceptible to clindamycin, vancomycin is the only validated option for intrapartum prophylaxis in patients with a penicillin allergy. GBS is the most common infectious cause of morbidity and mortality in neonates [8]. The primary risk factor for early-onset GBS infection in neonates is colonization of the maternal rectum or genital tract, and between 10–30% of women are colonized with GBS [9–12]. Because of the significant morbidity and mortality associated with neonatal GBS infection, both the Centers for Disease Control and Prevention and American College of Obstetricians and Gynecologists (ACOG) recommend universal culture-based GBS screening for all prenatal patients between 36 0/7 and 37 6/7 weeks of gestation [13]. GBS remains susceptible to both penicillin and ampicillin, and penicillin is the treatment of choice for GBS infections and prophylaxis because of its low cost, effectiveness, and narrow antimicrobial spectrum [8]. For this reason, ACOG has recommended penicillin allergy testing for pregnant women who report a penicillin or amoxicillin allergy [13]. This recommendation aligns with the American Academy of Allergy, Asthma and Immunology's 2016 recommendation that penicillin allergy testing should be routinely performed on all patients with a listed allergy to penicillin, ampicillin, or amoxicillin [14]. Using economic modeling, Sousa-Pinto *et al*. evaluated penicillin allergy testing techniques to evaluate cost-savings in 24 decision models [15]. They found penicillin allergy testing to be cost saving, with a net benefit ranging from $256 to $6745 for inpatients and outpatients, respectively. Protocols for penicillin allergy testing are well established, the cost of testing has been previously characterized in detail [16], and testing has been shown to be safe for pregnant women [17–19]. Considering pregnancy care and the delivery hospitalization, including GBS antimicrobial susceptibility testing, we compared the cost of usual care to the cost of a test then treat strategy for all pregnant women who report a penicillin allergy.

## Methods

We report our methods using the Consolidated Health Economic Evaluation Reporting Standards 2022 (CHEERS 2022) guidelines. In this institutional review board exempt study, we used billing data from a tertiary care center and previously published literature. Our decision tree compares the additional cost of testing for penicillin allergy to no test (usual care), which ensures appropriate antibiotic use for all women in the test then treat strategy. Our population of interest includes all pregnant women who report a penicillin allergy. We excluded outcomes for neonatal care and/or maternal post-cesarean surgical site infections.

### Decision tree model

To estimate the cost-effectiveness of a test then treat strategy for pregnant women who self-report a penicillin allergy, we constructed a decision tree. The decision tree compared the test then treat strategy to usual care. We selected appropriate or inappropriate antibiotic use as the endpoint of the model. Because of lack of published differential effectiveness outcomes for penicillin vs. alternative antibiotics, our model does not include neonatal or maternal post-cesarean surgical site infection outcomes. A diagram of our decision tree is shown in Fig 1.

Briefly, women under the test then treat strategy are first tested for a penicillin allergy and then treated according to their test results, while those under usual care are treated as if their self-reported allergy is confirmed. No antibiotics are routinely indicated for GBS negative women who give birth vaginally, but penicillin prophylaxis is indicated for those who are colonized with GBS. The second line antibiotic for GBS prophylaxis is clindamycin, with vancomycin reserved for those with clindamycin-resistant bacterial isolates. Therefore, women who have a penicillin allergy and whose rectal-vaginal cultures are positive for GBS should have additional testing of GBS isolates for clindamycin susceptibility. In those GBS positive patients attempting vaginal birth, vancomycin is prescribed when the GBS isolate is not susceptible to clindamycin. Alternately, for patients with no penicillin allergy, cefazolin is prescribed when giving birth via cesarean (planned or unplanned), regardless of GBS status. For those with a low-risk penicillin allergy, cefazolin may still be used at cesarean delivery, but for those with a history of anaphylaxis to beta lactams, clindamycin and gentamycin are used for surgical prophylaxis [20].

### Model inputs

Our model included the probability of true penicillin allergy, prevalence of GBS and clindamycin resistance, and rates of delivery modes (planned and unplanned cesarean, and vaginal). We searched previously published literature to identify these inputs. We assumed that the true rate of penicillin allergy among pregnant women was similar to what is reported among the general population (2%) [17]. GBS prevalence was estimated using a cohort study of all pregnant women who delivered at a Duke Health affiliated hospital between January 1, 2003 and December 31, 2015. That study found a GBS prevalence of 21.6% [1]. Clindamycin resistance was estimated from a study that collected two hundred random GBS single-patient isolates over a four year time period. That study found that 33% of samples had clindamycin resistance [21]. The rate of planned [17.4%] and unplanned cesarean deliveries (13.1%) were informed by The Consortium on Safe Labor, which used electronic medical records from 19 hospitals across the U.S. to approximate almost 4 million cesarean births [22].

Costs included the cost of antibiotics and testing. We report costs in 2020 U.S. dollars (USD) and we retrieved cost inputs from three major sources: billing data from our standardized cost data warehouse; previously published literature; and the Centers for Medicare and Medicaid Services fee schedule. We pulled the cost of each antibiotic by searching our

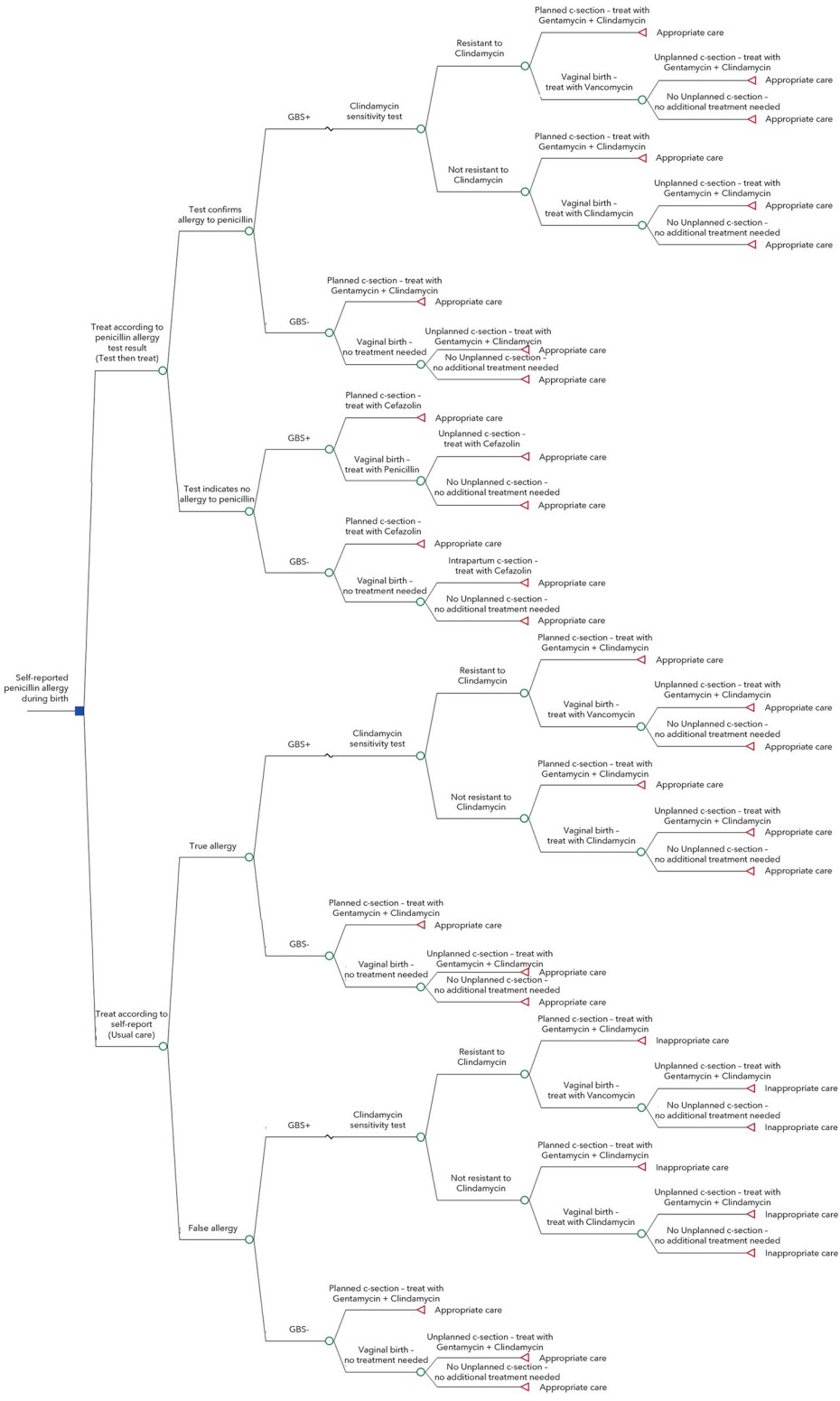

**Fig 1. Decision tree.**

**Table 1. Model inputs.**

|  | Mean | SD | Distribution |
|---|---|---|---|
| **Probabilities** |  |  |  |
| True penicillin allergy [17] | 0.06 | $7 \times 10^{-6}$ | Beta |
| Group B Streptococcus [1] | 0.216 | $2 \times 10^{-3}$ | Beta |
| Clindamycin resistant [21] | 0.33 | $3 \times 10^{-2}$ | Beta |
| Planned cesarean [22] | 0.174 | $2 \times 10^{-4}$ | Beta |
| Unplanned cesarean [22] | 0.131 | $2 \times 10^{-4}$ | Beta |
| **Per person costs, 2020 USD** |  |  |  |
| Cefazolin | $ 96 | $ 10 | Gamma |
| Clindamycin | $ 89 | $ 22 | Gamma |
| Gentamycin | $ 56 | $ 10 | Gamma |
| Penicillin | $ 65 | $ 19 | Gamma |
| Vancomycin | $ 68 | $ 10 | Gamma |
| Penicillin allergy test [16] | $ 256 | $ 144 | Gamma |
| Clindamycin resistant test [25] | $ 9 | $ 6 | Gamma |

standardized cost data warehouse for pregnant women giving birth between December 2017 to March 2021, who were prescribed any of the antibiotics of interest. The cost data warehouse is an internal resource that uses widely accepted methods to convert internal costs to standardized costs for publication [23]. To briefly describe the method, we use a hybrid model where Medicare reimbursement rates are applied to professional services. The hospital services are calculated by applying hospital cost-to-charge ratios to the charges. We report the mean and SD of each antibiotic, separately (Table 1). We included costs for all women and note that there was substantial variability due to differences among delivery times, which in turn affected the amount of antibiotics needed. Using those methods, we found that the average cost of a course of penicillin was $65; cefazolin $96; clindamycin $89; gentamycin $56; and vancomycin $68. The cost of a penicillin allergy test was informed by a cost study that included the following components: penicillin allergy evaluation, penicillin skin testing, and a 1-step amoxicillin drug challenge. It found that the average cost of a penicillin allergy test was $220 (2016 USD) [16]. We updated this cost to 2020 USD using the U.S. Bureau of Labor Statistics' Consumer Price Index for hospital services [24]. The estimated cost of penicillin allergy testing used in this analysis is probably higher than the real world, because many low-risk penicillin "allergic" individuals are currently delabeled by a direct oral amoxicillin challenge without antecedent skin testing. The cost of GBS sensitivity testing was informed by the Centers for Medicare and Medicaid Services fee schedule for procedure code 87186, which was listed as $9 (2020 USD) [25].

## Cost-effectiveness analysis

We conducted our cost-effectiveness analysis in TreeAge Pro v.2019. Our timeline included testing for penicillin allergy and GBS sensitivity, and delivery. Due to the short duration of our model timeline, no discounting of outcomes was needed. Our outcomes of interest included the total cost per patient and the proportion of patients with appropriate antibiotic use, under each strategy. To estimate the incremental difference between the two strategies of interest, we calculated a ratio comprising of the additional cost of the test then treat strategy over usual care, divided by the percent change in appropriate antibiotic prescription of the test then treat strategy over usual care. This incremental ratio may be interpreted as the average additional

cost per pregnant woman to ensure that they receive appropriate antibiotics during delivery; and this additional cost would only apply to women with a history of penicillin allergy.

## Sensitivity analysis

We conducted one-way and probabilistic sensitivity analysis (PSA). In the one-way sensitivity analysis, we varied each parameter one at a time by one standard deviation of its mean to identify the additional cost of test then treat compared to usual care. Our results are displayed in a tornado diagram (Fig 2A and 2B). For the PSA, we varied all inputs simultaneously in over

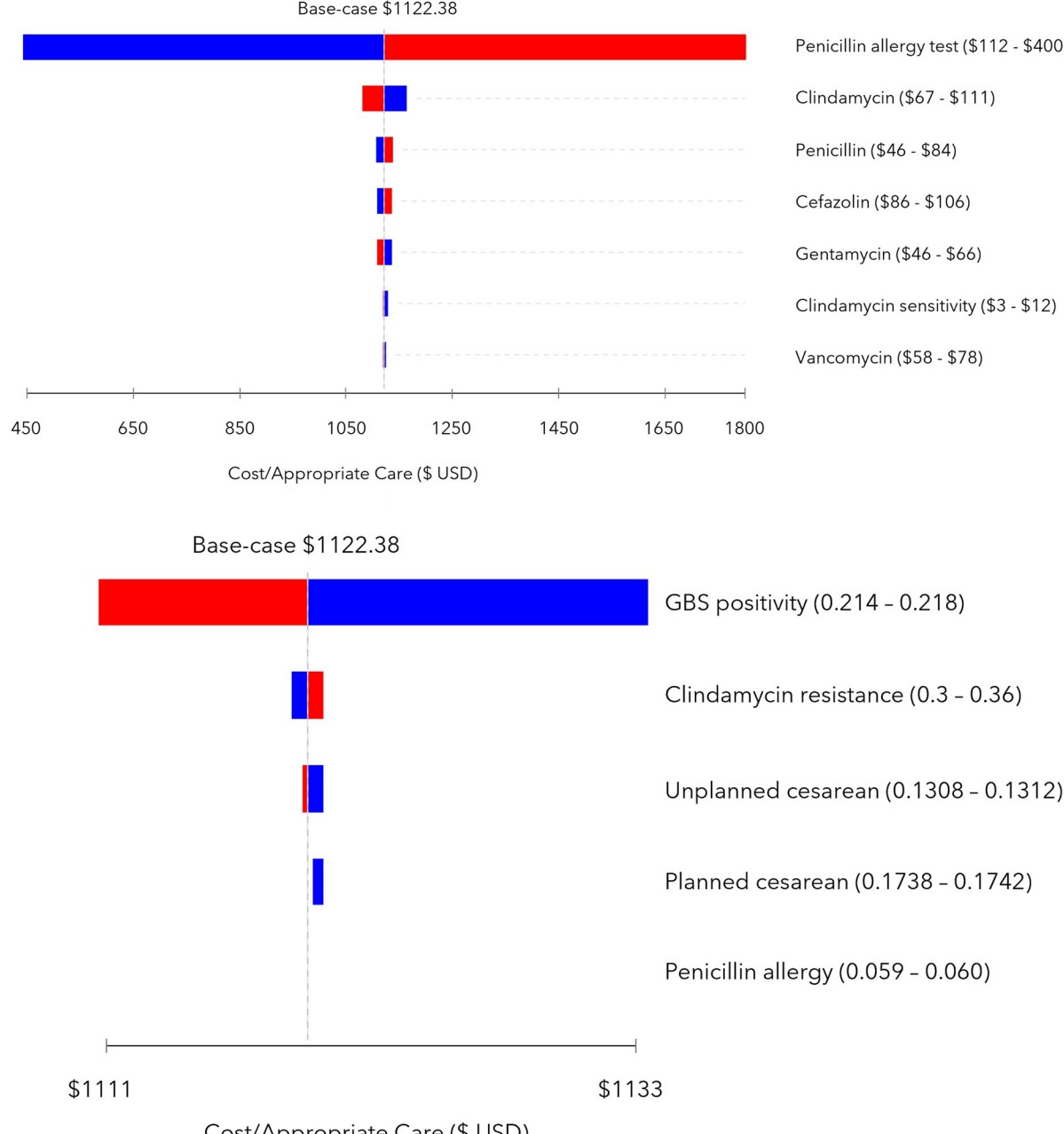

**Fig 2.** A. 1-way sensitivity analysis on cost parameters, tornado diagram. B. 1-way sensitivity analysis on probability parameters, tornado diagram.

10,000 simulations of the model. The inputs were sampled from the corresponding parameter distributions, using the means and standard deviations reported in Table 1. Probabilities were sampled from $\beta$ (beta) distributions and parameterized using data reported in cited source materials. Cost parameters were sampled from $\gamma$ (gamma) distributions and parameterized by referring to our internal data when available, or cited source materials when available. We also used tracker variables to identify the proportion of women who would receive appropriate and inappropriate antibiotic care under each treatment strategy. Results from the PSA are displayed as a scatter plot and as counts.

## Results

### Cost-effectiveness

We found that on average the cost of usual care was $57.55 per person and ensured appropriate antibiotic therapy for 79% of pregnant women with a self-reported penicillin allergy. The test then treat strategy was on average $295.15 per person and ensured that 100% of all pregnant women with a self-reported penicillin allergy would receive appropriate antibiotic therapy. Therefore, the additional cost to ensure appropriate GBS prophylaxis for all women who report a penicillin allergy, using the test then treat strategy, was $1122.38 per person reporting an allergy (Table 2).

### Sensitivity analysis

The cost parameter that had the greatest impact on our model was the cost of the penicillin allergy test (Fig 2A). From the one-way sensitivity analysis we found that if the cost of the penicillin allergy test were $112 (base-case $256), then the additional cost to ensure appropriate care for all women who report a penicillin allergy was $441.89 per person. Alternately, if the cost of the penicillin allergy test were $400, then the additional cost to ensure appropriate care for all women who report a penicillin allergy was $1802.40 per person. Varying the costs of the antibiotics only increased or decreased the cost of test then treat compared to usual care by $40 at most.

Varying the probabilities of GBS positivity, clindamycin resistance, cesarean birth, and positive penicillin allergy tests did not substantially affect our model results (Fig 2B). When we varied these probabilities, the cost of test then treat compared to usual care increased or decreased by no more than $10.

The incremental cost-effectiveness scatterplot is shown in Fig 3. Each dot represents one simulated scenario while the dark triangle represents our base-case incremental cost-effectiveness ratio (described previously as the cost to ensure appropriate care for all women who report a penicillin allergy, under the test then treat strategy, compared to usual care). The confidence ellipse contains 95% of all simulated scenarios. The upper right quadrant (quadrant I) contains estimates that are potentially cost-effective; however, it should be noted that because we did not specify how much more we would be willing to pay to ensure appropriate antibiotic use, it is not possible to say if test then treat is cost-effective. The lower right quadrant (quadrant IV) contains estimates that are cost-saving, as these estimates represent situations when test then treat is not only less costly than usual care but also increased appropriate antibiotic

**Table 2. Cost-effectiveness results.**

| | Cost | Δ Cost | Effect | Δ Effect | Cost/Effect | N appropriate antibiotic use | N in-appropriate antibiotic use |
|---|---|---|---|---|---|---|---|
| **Usual care** | $ 57.55 | | 0.79 | | | 7,843 | 2,157 |
| **Test then treat** | $ 295.15 | $ 237.59 | 1 | 0.21 | $ 1122.38 | 10,000 | 0 |

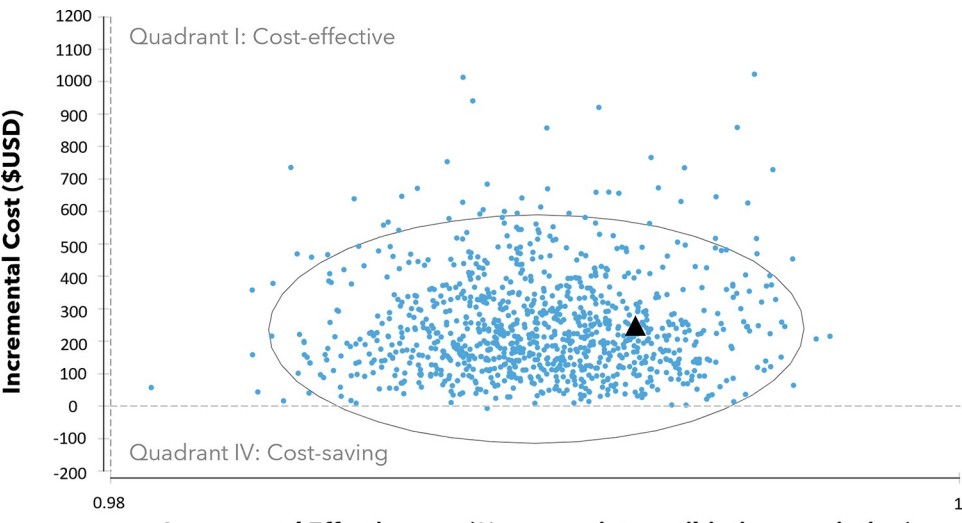

**Fig 3. Incremental cost-effectiveness scatterplot.**

use. Of the 10,000 simulated scenarios comparing test then treat to usual care, 99% resulted in potential cost-effectiveness (i.e., greater effectiveness at a higher cost) and 1% in cost-savings (i.e., a greater effectiveness at a lower cost). Our tracker variables showed that adopting a test than treat strategy in comparison to usual care would increase the number of appropriate antibiotic use from 7,843 to 10,000 and decrease the number of inappropriate antibiotic use from 2,157 to 0 (Table 2).

## Discussion

We evaluated the cost-effectiveness of routine penicillin allergy skin testing in pregnant women who report a penicillin allergy and found the mean additional cost of the test then treat strategy was $1122.38 per person reporting an allergy. This is a potentially cost-effective strategy and future research can explore this by incorporating long-term outcomes, including appropriate antibiotic therapy's effect on quality of life for mother and baby. It would be less costly to test for a penicillin allergy only among women who are GBS positive, but for timing purposes allergy testing usually occurs prior to GBS testing in clinical practice. Still, our model suggests that systematic prenatal penicillin allergy testing could reduce or eliminate inappropriate antimicrobial therapy for GBS prophylaxis. A retrospective study of pregnant women with reported penicillin allergy found women who received a penicillin allergy skin test had increased use of first-line antibiotics for GBS prophylaxis with intrapartum penicillin (adjusted odds ratio 26.9; 95% CI 6.32 to 114) and with cefazolin for cesarean delivery prophylaxis (adjusted odds ratio 1.94, 95% CI 1.06 to 3.52) compared to women who did not undergo allergy testing [26]. In addition to eliminating the need to use alternatives to penicillin for GBS prophylaxis in pregnancy, penicillin allergy testing may provide long-term benefit for women's future health care management, including during subsequent pregnancies.

The broader societal impact of de-labeling patients with penicillin allergy could potentially allow for future narrow-spectrum antibiotic selection and slow the growth of antibiotic-resistant bacteria [27]. There are significant public health implications associated with the unnecessary use of vancomycin including the emergence of vancomycin-resistant enterococci and

other resistant organisms. In our model, we were able to ensure that 100% of women who had penicillin allergy testing received the appropriate antibiotics during labor and delivery, therefore limiting unnecessary use of alternative antibiotics. We did not account for other indications for antibiotic use during pregnancy, including urinary tract infections, which may benefit from use of a beta lactam.

Strengths of our study include the use of a robust decision tree and modeling based on actual costs of interventions and medications. The sensitivity analysis indicates large cost variation with only one component of the model, the cost of penicillin allergy testing, so the results are likely to hold true across settings. The model included patients undergoing vaginal birth, planned cesarean, and unplanned cesarean birth, thus accounting for every mode of delivery and multiple appropriate antibiotic regimens. Some people may encounter barriers to obtaining testing. The extra time and cost needed for an allergy testing visit may be a deterrent for some pregnant people [17]. In addition, rural patients may not have access to a clinic that offers allergy testing. Undergoing testing may also create anxiety in people who are fearful of an adverse event, and similarly some health systems may not be able to accommodate the increased volume of testing resulting from the ACOG recommendation.

As a limitation, our analysis may under-estimate the cost-effectiveness of the test then treat intervention because it does not include cost estimates for neonatal care or for maternal post-delivery care, and because we considered only the index pregnancy in our model. For neonatal cost considerations, guidance from the Centers for Disease Control and Prevention does not include alternatives to penicillin or cephalosporins as options for achieving adequate neonatal prophylaxis, so despite prophylaxis, infants of GBS positive mothers who received clindamycin or vancomycin are often subjected to further in-hospital testing and observation compared to those with penicillin treatment [28]. Because it will increase the number of neonates receiving adequate prophylaxis during delivery, penicillin allergy testing is likely to decrease the cost of immediate neonatal care. We also anticipate that post-cesarean wound infection frequency and the associated cost of that complication may be higher in those patients who do not receive the most appropriate antibiotics [29]. Results are reported as incremental cost of the intervention rather than in quality-adjusted life years (QALYs) because the model does not include neonatal lives saved or other costs beyond intrapartum antibiotic prophylaxis. Finally, because sensitivity and specificity of pencillin allergy testing have not been definitively determined, our model is limited by assuming 100% sensitivity and not accounting for the potential high cost of false negative testing followed by anaphylactic reaction.

## Conclusion

ACOG states penicillin allergy testing is safe during pregnancy and recommends people with a reported penicillin allergy undergo testing, if available. The test then treat strategy is a good-value intervention to ensure appropriate antibiotic use for the current delivery. Additional work is needed to evaluate the cost benefit of testing for additional pregnancies and for future non-pregnancy healthcare of the mother.

## Author Contributions

**Conceptualization:** Emily E. Sharpe, Bijan J. Borah, Margaret C. Gill, Regan N. Theiler.

**Data curation:** Ruchita Dholakia, Hannah H. Ahn, James P. Moriarty.

**Formal analysis:** Viengneesee Thao, James P. Moriarty, Bijan J. Borah.

**Funding acquisition:** Margaret C. Gill, Regan N. Theiler.

**Investigation:** Hannah H. Ahn.

**Methodology:** Viengneesee Thao, James P. Moriarty, Bijan J. Borah, Regan N. Theiler.

**Software:** Ruchita Dholakia.

**Supervision:** Margaret C. Gill, Regan N. Theiler.

**Writing – original draft:** Viengneesee Thao.

**Writing – review & editing:** Emily E. Sharpe, Hannah H. Ahn, James P. Moriarty, Bijan J. Borah, Margaret C. Gill, Regan N. Theiler.

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
