## [Decision Letter · Decision Letter 0]

6 Sep 2022

PONE-D-22-17964EVALUATING THE COST-EFFECTIVENESS OF TESTING PREGNANT WOMEN FOR PENICILLIN ALLERGYPLOS ONE

Dear Dr. Theiler,

Thank you for submitting your manuscript to PLOS ONE. After careful consideration, we feel that it has merit but does not fully meet PLOS ONE’s publication criteria as it currently stands. Therefore, we invite you to submit a revised version of the manuscript that addresses the points raised during the review process.The authors should minimize the length of the manuscript, in particular the introduction section and revise as per the journal guideline.The authors also must provide a clear justification about the overestimation of cost-effectiveness of the intervention, as it more deviated from real scenario.  They also critically elaborate  the target population to which the finding is inferred and assumptions used to evaluate their objective.Which guideline was applied to design the research protocol.They have to address  critical comments raised by the reviewers, specially the value addition of the research for clinical practice/inputs and alternative medicines that may affect the current estimated figure.  Please submit your revised manuscript by Oct 21 2022 11:59PM.  If you will need more time than this to complete your revisions, please reply to this message or contact the journal office at plosone@plos.org. Please include the following items when submitting your revised manuscript:A rebuttal letter that responds to each point raised by the academic editor and reviewer(s). You should upload this letter as a separate file labeled 'Response to Reviewers'.A marked-up copy of your manuscript that highlights changes made to the original version. You should upload this as a separate file labeled 'Revised Manuscript with Track Changes'.An unmarked version of your revised paper without tracked changes. You should upload this as a separate file labeled 'Manuscript'.

We look forward to receiving your revised manuscript.

Kind regards,

Tefera Chane Mekonnen, Master in Public Health(MPH)

Academic Editor

PLOS ONE

Journal Requirements:

“This work was supported by the Advance the Practice Research Award from the Mayo Clinic Office of Translation to Practice. The content is solely the responsibility of the authors and does not necessarily represent the official views of Mayo Clinic.”

“MG and RT received competitive funding from the Mayo Clinic Office of Translation to Practice. The funders had no role in study design, data collection and analysis, decision to publish, or preparation of the manuscript.”

“Dr. Theiler has a know-how agreement and research funding from HeraMed and serves on the Medical Advisory Board for Delfina Care. Dr. Borah is a consultant to Exact Sciences and Boehringer-Ingelheim on unrelated (non-Ob/Gyn) projects.”

Reviewers' comments:

Reviewer's Responses to Questions

**Comments to the Author**

1. Is the manuscript technically sound, and do the data support the conclusions?

Reviewer #1: No

Reviewer #2: No

Reviewer #3: Yes

2. Has the statistical analysis been performed appropriately and rigorously? 

Reviewer #1: No

Reviewer #2: N/A

Reviewer #3: Yes

3. Have the authors made all data underlying the findings in their manuscript fully available?

Reviewer #1: Yes

Reviewer #2: Yes

Reviewer #3: No

4. Is the manuscript presented in an intelligible fashion and written in standard English?

Reviewer #1: Yes

Reviewer #2: Yes

Reviewer #3: Yes

5. Review Comments to the Author

Reviewer #1: The authors present a cost-effectiveness analysis of performing penicillin allergy testing in pregnancy on women that ultimately test positive for GBS.

Introduction

1. The authors need to reduce this section by half. Focus more on the effect of penicillin allergy testing in pregnancy. For example, delete lines 83 to 98.

2. Line 98 to 100 – The objective should be clarified to indicate in pregnant individuals that ultimately test positive for GBS colonization.

Methods

3. Did the authors follow the CHEERS guidelines (Husereau et al. CHEERS 2022 ISPOR Good Research Practices Task Force. Consolidated Health Economic Evaluation Reporting Standards 2022 (CHEERS 2022) Statement: Updated Reporting Guidance for Health Economic Evaluations) for this CEA? If so, please state this.

4. Lines 103 to 105 – It was not clear until you read the Discussion that this CEA was excluding outcomes for neonatal care and/or maternal post-cesarean surgical site infections. This exclusion should be stated up front.

5. Lines 107 to 108 – The authors should better define the population that will be tested. A suggestion would be to use the National Vital Statistics Reports for Births: Final data for 2020. You could then take the 3,613,647 births in 2020 and multiply this number by the estimated rate of individuals who report a penicillin allergy during pregnancy. This would be the cohort to then perform cost estimates on. Further a more robust estimate of the rate of cesarean delivery could be estimated. [Osterman et al. Births: Final data for 2020. National Vital Statistics Reports; vol 70 no 17. Hyattsville, MD: National Center for Health Statistics. 2022].

6. Line 110, Figure 1 - The decision tree is lacking an important outcome option. When a woman reports a penicillin allergy, the recommended antibiotic for intrapartum antibiotic prophylaxis, if she is colonized with GBS, is based on her risk of a severe reaction and the susceptibility of the GBS isolate to clindamycin. For women whose reported penicillin allergy indicates a low risk of anaphylaxis or uncertain severity, a first-generation cephalosporin (ie, cefazolin) is recommended. Many institutions do not perform sensitivity to GBS isolates in this clinical situation and reserve sensitivity testing to clindamycin for individuals that report a high risk of anaphylaxis. In addition, there are estimates available in the medical literature that show the rate of reported low versus high risk of anaphylaxis to penicillin in pregnancy. This treatment option needs to be added to the algorithm.

7. Lines 129 to 131 – A more robust range of GBS maternal colonization in pregnancy in the US (22% to 28%) is available (Russell et al. Maternal colonization with Group B Streptococcus and serotype distribution worldwide: systematic review and meta-analysis. Clin Infect Dis 2017; 65 (Suppl 2): S100-S11).

8. Lines 137 to 150 – The costs associated with the antibiotics are for how many doses? For example, 56% of pregnant individuals will receive two or more doses of intrapartum penicillin for maternal GBS prophylaxis. In addition, the recommended dose and administration interval of vancomycin for maternal GBS prophylaxis changed in 2019. Does this price estimate reflect that?

9. Lines 151 to 153 – The authors cite the Blumenthal article from 2018 (reference 14) that reported a cost for a one-step penicillin allergy testing. Yet the Desravines article from 2021 recommends a 3 step testing (skin prick, intradermal testing, oral challenge) [reference 20]. Should this increased costs have been estimated in the model?

Reviewer #2: General comments:

It has been noted that the greatest reductions in healthcare costs associated with penicillin allergy delabeling are due to less urgent care and outpatient utilization and shorter durations of hospitalizations. It does not appear that these potential effects were incorporated into your modeling.

https://pubmed.ncbi.nlm.nih.gov/28366717/

Please comment on how removing all warnings not to use cephalosporins in the setting of a penicillin allergy would affect your modeling.

https://pubmed.ncbi.nlm.nih.gov/33914051/

The reported rate of unconfirmed penicillin allergy is about 8.0% in pregnant women and up to about 7% are confirmed if all are skin tested prior to an oral amoxicillin challenge. Thus, the true rate of penicillin allergy in pregnant women is only about 0.5%. Women with a penicillin allergy, with or without GBS, also had significantly (about 10%) higher cesarean section rates and spent significantly more (about 0.1) days in the hospital after delivery. Please consider incorporating these data into your modeling.

https://pubmed.ncbi.nlm.nih.gov/33278285/

https://pubmed.ncbi.nlm.nih.gov/28333608/

Specific comments:

Line 153: Please consider commenting here that the estimated cost of penicillin allergy testing used in this analysis is probably higher than the real world, because many low-risk penicillin “allergic” individuals are currently delabeled by a direct oral amoxicillin challenge without antecedent skin testing.

Reviewer #3: This is a straightforward economic evaluation of the cost of introducing testing for penicillin allergy amongst pregnant women and people who believe or have been labelled as being allergic. This is within the context of universal antenatal screening for GBS and use of penicillin intrapartum antibiotic prophylaxis for those testing positive for GBS colonisation.

I could not find the data in Blumthenthal 2018 that suggests those with penicillin allergies are a higher (50%) the risk of surgical site infections. Could this have been the wrong reference?

The model inputs and costs seem appropriately obtained/ derived and used in analysis. I am not familiar with the way in which US healthcare costs are warehoused and how Medicare/Medicaid tariffs are used. The authors state that this information cannot be made publicly available.

Figure 1 model – either strategy, GBS+, resistant to clindamycin: need to explain why for planned/ unplanned C-section the path include clindamycin (alongside gentamycin)? Presumably for maternal prophylaxis for endometritis and not for prevention of early onset neonatal GBS infection.

The outcome is incremental cost per additional women receiving appropriate antibiotics. This is appropriate for short-term modelling – ultimately it the impact on neonatal GBS infection that is most important.

Is it appropriate to assume that the penicillin allergy test is 100% accurate? I appreciate adding in false positive/ negatives expands the model, but there could be substantial costs associated with anaphylaxis from a false negative result.

The absence of a willingness to pay threshold set by policy makers hinders interpretation, as does a utility such as a QALY that can be used to compare against other strategies.

The results pertain only to those women who report or believe they have a penicillin allergy based on past experience. There will be women who have never been exposed to penicillin who will be offered this antibiotic if GBS positive. Could the model be expanded to include women of unknown penicillin allergy status, and also both models restricted to testing only the positive for GBS colonisation. I appreciate that this would need to occur a few days after the GBS test, will probably involve another appointment and may not occur before birth, so have no impact on care. The timepoint at which the allergy test would occur during pregnancy is not stated, and could itself mean another appointment. The authors discuss the societal cost and limitations that an allergy test might bring.

I am not sure of the policy of PLOSOne but generally in the OBGYN literature, we refer to pregnant women or pregnant people, not pregnant patients.

Overall, an interesting and valuable piece of research, albeit limited in generalisability.

6. PLOS authors have the option to publish the peer review history of their article (what does this mean?). If published, this will include your full peer review and any attached files.

Reviewer #1: No

Reviewer #2: **Yes: **Eric Macy MD MS FAAAAI

Reviewer #3: **Yes: **Prof Jane Daniels

---

## [Author Response · Author response to Decision Letter 0]

12 Oct 2022

General comments:

The authors should minimize the length of the manuscript, in particular the introduction section and revise as per the journal guideline.

Thank you for this suggestion. Per reviewer 1’s suggestions, we have omitted lines 90-104 (originally lines 83 to 98).

The authors also must provide a clear justification about the overestimation of cost-effectiveness of the intervention, as it more deviated from real scenario. They also critically elaborate the target population to which the finding is inferred and assumptions used to evaluate their objective.

 The work estimates the cost per penicillin allergic patient at the onset of care, which in an ideal scenario approximates the cost per appropriate dose of antibiotic because of the small number of true allergies in this population. Given that uptake of testing will not be 100%, our numbers represent the ideal state and the actual cost in practice will be lower because of the difference in uptake of testing. Additionally, we have modeled the more conservative protocol of skin testing followed by oral challenge, and oral challenge alone would be less costly. We have added a sentence (lines 176-179) reflecting this difference in testing protocols.

Which guideline was applied to design the research protocol.

We followed the guidelines published by the 2nd Panel on Cost-Effectiveness in Health and Medicine (PMID: 27623463 DOI: 10.1001/jama.2016.12195).

They have to address critical comments raised by the reviewers, specially the value addition of the research for clinical practice/inputs and alternative medicines that may affect the current estimated figure.

We have responded below to the reviewer comments.

Editor comments:

and

Thank you for these templates, we have updated our manuscript to abide by PLOS One’s style requirements. All changes are tracked in our updated manuscript.

Please include your full ethics statement in the ‘Methods’ section of your manuscript file. In your statement, please include the full name of the IRB or ethics committee who approved or waived your study, as well as whether or not you obtained informed written or verbal consent. If consent was waived for your study, please include this information in your statement as well.

Our study was exempt from IRB review and we have updated our Methods section to reflect this in the following way: In this institutional review board exempt study, we used billing data from a tertiary care center and previously published literature, (lines 115 to 116) in our track changes manuscript.

Thank you for stating the following in the Acknowledgments Section of your manuscript:

“This work was supported by the Advance the Practice Research Award from the Mayo Clinic Office of Translation to Practice. The content is solely the responsibility of the authors and does not necessarily represent the official views of Mayo Clinic.”

“MG and RT received competitive funding from the Mayo Clinic Office of Translation to Practice. The funders had no role in study design, data collection and analysis, decision to publish, or preparation of the manuscript.”

We have removed our funding statement from our manuscript. Please update our funding statement to the following: 

MG and RT received competitive funding from the Mayo Clinic Office of Translation to Practice via the Advance the Practice Research Award. The funders had no role in study design, data collection and analysis, decision to publish, or preparation of the manuscript.

Thank you for stating the following in the Competing Interests section:

“Dr. Theiler has a know-how agreement and research funding from HeraMed and serves on the Medical Advisory Board for Delfina Care. Dr. Borah is a consultant to Exact Sciences and Boehringer-Ingelheim on unrelated (non-Ob/Gyn) projects.”

We have added the requested financial disclosure statement and disclosed our competing interests in the revised cover letter. 

In your Data Availability statement, you have not specified where the minimal data set underlying the results described in your manuscript can be found. PLOS defines a study's minimal data set as the underlying data used to reach the conclusions drawn in the manuscript and any additional data required to replicate the reported study findings in their entirety. All PLOS journals require that the minimal data set be made fully available. For more information about our data policy, please see http://journals.plos.org/plosone/s/data-availability.

Upon re-submitting your revised manuscript, please upload your study’s minimal underlying data set as either Supporting Information files or to a stable, public repository and include the relevant URLs, DOIs, or accession numbers within your revised cover letter. For a list of acceptable repositories, please see http://journals.plos.org/plosone/s/data-availability#loc-recommended-repositories. Any potentially identifying patient information must be fully anonymized. Important: If there are ethical or legal restrictions to sharing your data publicly, please explain these restrictions in detail. Please see our guidelines for more information on what we consider unacceptable restrictions to publicly sharing data:

Thank you. Our decision tree model can be easily replicated by using the figures we listed in Table 1. Most of the data used in our model is available publically and we have cited those sources accordingly. The only data not publically available is the cost of antibiotics data which was pulled from patients admitted to Mayo Clinic in Rochester, MN. The cost data are stored on safe servers at Mayo Clinic, USA and handled confidentially. If access to this data is needed, application for access may be made to the Economic Evaluation Service at the Kern Center for the Science of Health Care Delivery (ees@mayo.edu). Upon approval, applicants would be able to access these data in the same manner as the authors.

http://journals.plos.org/plosone/s/data-availability#loc-unacceptable-data-access-restrictions. Note that it is not acceptable for the authors to be the sole named individuals responsible for ensuring data access.

The cost of antibiotics data used to estimate the means and standard deviations in Table 1 is stored by the research group of authors on safe servers at Mayo Clinic. Please refer to our statement about data sharing. All other data has been published and is publically available. 

Reviewer 1 comments:

Introduction

The authors need to reduce this section by half. Focus more on the effect of penicillin allergy testing in pregnancy. For example, delete lines 83 to 98.

We appreciate this suggestion and have deleted lines 83 to 98.

Line 98 to 100 – The objective should be clarified to indicate in pregnant individuals that ultimately test positive for GBS colonization.

Thank you. Line 112 has been updated to reflect that we modeled testing of ALL pregnant patients reporting penicillin allergy. Because of the cadence of prenatal care, with GBS testing at the end of the pregnancy, our decision tree reflects testing regardless of subsequent GBS culture results. 

Methods

Did the authors follow the CHEERS guidelines (Husereau et al. CHEERS 2022 ISPOR Good Research Practices Task Force. Consolidated Health Economic Evaluation Reporting Standards 2022 (CHEERS 2022) Statement: Updated Reporting Guidance for Health Economic Evaluations) for this CEA? If so, please state this.

Yes, we did follow the CHEERS guideline for reporting and we have added the following statement to our methods section (lines 114 to115): We report our methods using the Consolidated Health Economic Evaluation Reporting Standards 2022 (CHEERS 2022) guidelines.

Lines 103 to 105 – It was not clear until you read the Discussion that this CEA was excluding outcomes for neonatal care and/or maternal post-cesarean surgical site infections. This exclusion should be stated up front.

Thank you for this recommendation. We have added the following to our methods section: “Because of lack of published differential effectiveness outcomes for penicillin vs. alternative antibiotics, our model does not include neonatal or maternal post-cesarean surgical site infection outcomes.” (lines 127-129). We have also acknowledged this limitation in the discussion section: “As a limitation, our analysis may under-estimate the cost-effectiveness of the test then treat intervention because it does not include cost estimates for neonatal care or for maternal post-delivery care, and because we considered only the index pregnancy in our model.”

Lines 107 to 108 – The authors should better define the population that will be tested. A suggestion would be to use the National Vital Statistics Reports for Births: Final data for 2020. You could then take the 3,613,647 births in 2020 and multiply this number by the estimated rate of individuals who report a penicillin allergy during pregnancy. This would be the cohort to then perform cost estimates on. Further a more robust estimate of the rate of cesarean delivery could be estimated. [Osterman et al. Births: Final data for 2020. National Vital Statistics Reports; vol 70 no 17. Hyattsville, MD: National Center for Health Statistics. 2022].

Thank you for this suggestion. The cesarean rate reported by Osterman et al. is the overall cesarean rate; however, for our paper we need the planned and unplanned cesarean rate. For this reason, we have opted to use the rates reported by Zhang et al. Furthermore, there is only a slight difference in total cesarean rates between the Osterman article (31.8%) and the Zhang article (30.5%), and this difference has been captured in our sensitivity analysis.

Line 110, Figure 1 – The decision tree is lacking an important outcome option. When a woman reports a penicillin allergy, the recommended antibiotic for intrapartum antibiotic prophylaxis, if she is colonized with GBS, is based on her risk of a severe reaction and the susceptibility of the GBS isolate to clindamycin. For women whose reported penicillin allergy indicates a low risk of anaphylaxis or uncertain severity, a first-generation cephalosporin (ie, cefazolin) is recommended. Many institutions do not perform sensitivity to GBS isolates in this clinical situation and reserve sensitivity testing to clindamycin for individuals that report a high risk of anaphylaxis. In addition, there are estimates available in the medical literature that show the rate of reported low versus high risk of anaphylaxis to penicillin in pregnancy. This treatment option needs to be added to the algorithm.

Thank you for pointing this out as an alternative. For our decision tree we have used the ACOG-recommended algorithm for GBS sensitivity testing and subsequent treatment.

Lines 129 to 131 – A more robust range of GBS maternal colonization in pregnancy in the US (22% to 28%) is available (. Clin Infect Dis 2017; 65 (Suppl 2): S100-S11).

Thank you for this reference. The above article actually cites an 18% estimate of maternal colonization with GBS. On our review of the literature with an emphasis on U.S. costs, we felt the estimate of 21.6% was most consistent with other U.S. publications and have chosen to base the analysis on that rate as cited. Ultimately, the prevalence of GBS had little impact on our cost outcome because of the sequence of care (GBS testing and sensitivity performed after allergy testing) and we do not think a small change in prevalence will meaningfully impact our primary outcome.

Lines 137 to 150 – The costs associated with the antibiotics are for how many doses? For example, 56% of pregnant individuals will receive two or more doses of intrapartum penicillin for maternal GBS prophylaxis. In addition, the recommended dose and administration interval of vancomycin for maternal GBS prophylaxis changed in 2019. Does this price estimate reflect that?

Thank you for this question. In our methods section we state that there was substantial variability in cost of antibiotics due to differences among labor and delivery times, which in turn affected number of doses of antibiotics needed. Thus, we report the average cost of each antibiotic in our model. The variation in cost and dosing thus reflects the variation in human labor duration and we do not specify a number of doses. The changing of dosage guidelines is reflected in our sensitivity analysis.

Lines 151 to 153 – The authors cite the Blumenthal article from 2018 (reference 14) that reported a cost for a one-step penicillin allergy testing. Yet the Desravines article from 2021 recommends a 3 step testing (skin prick, intradermal testing, oral challenge) [reference 20]. Should this increased costs have been estimated in the model?

Thank you. The Blumenthal article reports on “penicillin skin testing” followed by 1-step oral amoxicillin challenge, which they describe as comprehensive. We assumed this to include skin prick followed by intradermal challenge, which is the standard of care. The “one step” refers to the oral amoxicillin challenge component of testing, and we believe their analysis to be the most comprehensive estimate of actual cost for testing.

Reviewer 2 comments:

It has been noted that the greatest reductions in healthcare costs associated with penicillin allergy delabeling are due to less urgent care and outpatient utilization and shorter durations of hospitalizations. It does not appear that these potential effects were incorporated into your modeling. https://pubmed.ncbi.nlm.nih.gov/28366717/

Thank you for this question. Our model only pertains to women giving birth in an inpatient hospital setting, therefore, patients from urgent care and outpatient settings would not have been included in our model. We do believe considerable cost savings may be achieved over the subsequent life years of these patients and have commented on that in the discussion.

Please comment on how removing all warnings not to use cephalosporins in the setting of a penicillin allergy would affect your modeling. https://pubmed.ncbi.nlm.nih.gov/33914051/

We have incorporated appropriate cephalosporin use according to ACOG clinical guidelines into our decision tree, and previous publications have done the same—including that referenced above as well as our own (https://pubmed.ncbi.nlm.nih.gov/35906008/). The reference cited above does not include a recommendation to remove warnings about cephalosporin use. Removal of all recommendations to avoid cephalosporins in penicillin allergic individuals would likely negate the need for nearly all penicillin allergy testing if we consider cefazolin to be the appropriate substitute in such patients, but we are not aware of such a recommendation for practice change so did not include this discussion in the in the manuscript. We have included a citation of the ACOG practice bulletin with recommendations for prophylactic antibiotics, citation 24 (Use of prophylactic antibiotics in labor and delivery. ACOG Practice Bulletin No. 199. American College of Obstetricians and Gynecologists. Obstet Gynecol 2018;132:e103–19.)

The reported rate of unconfirmed penicillin allergy is about 8.0% in pregnant women and up to about 7% are confirmed if all are skin tested prior to an oral amoxicillin challenge. Thus, the true rate of penicillin allergy in pregnant women is only about 0.5%. Women with a penicillin allergy, with or without GBS, also had significantly (about 10%) higher cesarean section rates and spent significantly more (about 0.1) days in the hospital after delivery. Please consider incorporating these data into your modeling. https://pubmed.ncbi.nlm.nih.gov/33278285/

https://pubmed.ncbi.nlm.nih.gov/28333608/

We agree that the literature suggests higher cesarean rates, and thus longer duration of inpatient stay, in patients with reported PCN allergy. Our decision tree model did not include this differential cesarean rate because we do not know whether patients with confirmed vs. reported allergies will differ in their rates of unplanned cesarean birth, so we felt the most conservative approach was to use overall population data. The above study cites patients with “unverified penicillin allergy”. Larger prospective studies of pregnancy outcomes after incorporation of penicillin allergy testing protocols will be needed prior to using the above data in modeling. 

Line 153: Please consider commenting here that the estimated cost of penicillin allergy testing used in this analysis is probably higher than the real world, because many low-risk penicillin “allergic” individuals are currently delabeled by a direct oral amoxicillin challenge without antecedent skin testing.

Thank you for this suggestion. We have included this exact statement in our methods section (lines 175 to178).

Reviewer 3 comments:

This is a straightforward economic evaluation of the cost of introducing testing for penicillin allergy amongst pregnant women and people who believe or have been labelled as being allergic. This is within the context of universal antenatal screening for GBS and use of penicillin intrapartum antibiotic prophylaxis for those testing positive for GBS colonisation.

I could not find the data in Blumthenthal 2018 that suggests those with penicillin allergies are a higher (50%) the risk of surgical site infections. Could this have been the wrong reference?

Thank you for catching the citation error. We have added the correct citation: https://pubmed.ncbi.nlm.nih.gov/35281850/

The model inputs and costs seem appropriately obtained/ derived and used in analysis. I am not familiar with the way in which US healthcare costs are warehoused and how Medicare/Medicaid tariffs are used. The authors state that this information cannot be made publicly available.

The cost of antibiotics data used to estimate the means and standard deviations in Table 1 is stored by the research group of authors on safe servers at Mayo Clinic. Please refer to our statement about data sharing. All other data has been published and is publically available. 

Figure 1 model – either strategy, GBS+, resistant to clindamycin: need to explain why for planned/ unplanned C-section the path include clindamycin (alongside gentamycin)? Presumably for maternal prophylaxis for endometritis and not for prevention of early onset neonatal GBS infection.

The model includes clindamycin for cesarean surgical site infection prophylaxis in accordance with ACOG guidelines, reference number 24. We have added this reference (Use of prophylactic antibiotics in labor and delivery. ACOG Practice Bulletin No. 199. American College of Obstetricians and Gynecologists. Obstet Gynecol 2018;132:e103–19.)

The outcome is incremental cost per additional women receiving appropriate antibiotics. This is appropriate for short-term modelling – ultimately it the impact on neonatal GBS infection that is most important.

We agree and have stated this as a limitation of the study. When data become available detailing the relative effectiveness of non-penicillin antibiotics for GBS prophlyaxis, we recommend updating the analysis to incorporate long-term neonatal outcomes. 

Is it appropriate to assume that the penicillin allergy test is 100% accurate? I appreciate adding in false positive/ negatives expands the model, but there could be substantial costs associated with anaphylaxis from a false negative result.

We agree that false negative testing in particular may add cost because of rare anaphylactic reactions. However, the exact sensitivity and specificity of penicillin allergy testing have not been well described in the literature to date so were not incorporated into our model. Our model thus assumes 100% sensitivity and may underestimate cost. We have added the following to the discussion, lines 308-311: Finally, because sensitivity and specificity of pencillin allergy testing have not been definitively determined, our model is limited by assuming 100% sensitivity and not accounting for the potential high cost of false negative testing followed by anaphylactic reaction.

The absence of a willingness to pay threshold set by policy makers hinders interpretation, as does a utility such as a QALY that can be used to compare against other strategies.

We agree that this is a limitation of the study as noted in the discussion section. We do believe that this addition to the literature, in combination with other studies, may help to develop the above thresholds in the future. 

The results pertain only to those women who report or believe they have a penicillin allergy based on past experience. There will be women who have never been exposed to penicillin who will be offered this antibiotic if GBS positive. Could the model be expanded to include women of unknown penicillin allergy status, and also both models restricted to testing only the positive for GBS colonisation. I appreciate that this would need to occur a few days after the GBS test, will probably involve another appointment and may not occur before birth, so have no impact on care. The timepoint at which the allergy test would occur during pregnancy is not stated, and could itself mean another appointment. The authors discuss the societal cost and limitations that an allergy test might bring.

Thank you for this suggestion. We have discussed the timing of testing prior to GBS culture and that impact on cost, lines 261-263. Because of the rarity of penicillin anaphylaxis in patients without previously described allergy, we did not include that potential cost in the model. 

I am not sure of the policy of PLOSOne but generally in the OBGYN literature, we refer to pregnant women or pregnant people, not pregnant patients.

Thank you for this suggestion. We have updated the language throughout our manuscript to remove pregnant ‘patients’.

---

## [Decision Letter · Decision Letter 1]

12 Dec 2022

PONE-D-22-17964R1EVALUATING THE COST-EFFECTIVENESS OF TESTING PREGNANT WOMEN FOR PENICILLIN ALLERGYPLOS ONE

Dear Dr. Theiler,

Thank you for submitting your manuscript to PLOS ONE. After careful consideration, we feel that it has merit but does not fully meet PLOS ONE’s publication criteria as it currently stands. Therefore, we invite you to submit a revised version of the manuscript that addresses the points raised during the review process.Address comments raised by reviewers.Please submit your revised manuscript by 26/12/22. If you will need more time than this to complete your revisions, please reply to this message or contact the journal office at plosone@plos.org. Please include the following items when submitting your revised manuscript:A rebuttal letter that responds to each point raised by the academic editor and reviewer(s). You should upload this letter as a separate file labeled 'Response to Reviewers'.A marked-up copy of your manuscript that highlights changes made to the original version. You should upload this as a separate file labeled 'Revised Manuscript with Track Changes'.An unmarked version of your revised paper without tracked changes. You should upload this as a separate file labeled 'Manuscript'.If applicable, we recommend that you deposit your laboratory protocols in protocols.io to enhance the reproducibility of your results. Protocols.io assigns your protocol its own identifier (DOI) so that it can be cited independently in the future. For instructions see: https://journals.plos.org/plosone/s/submission-guidelines#loc-laboratory-protocols. Additionally, PLOS ONE offers an option for publishing peer-reviewed Lab Protocol articles, which describe protocols hosted on protocols.io. Read more information on sharing protocols at https://plos.org/protocols?utm_medium=editorial-email&utm_source=authorletters&utm_campaign=protocols.

We look forward to receiving your revised manuscript.

Kind regards,

Tefera Chane Mekonnen, Master in Public Health(MPH)

Academic Editor

PLOS ONE

Journal Requirements:

Reviewers' comments:

Reviewer's Responses to Questions

**Comments to the Author**

1. If the authors have adequately addressed your comments raised in a previous round of review and you feel that this manuscript is now acceptable for publication, you may indicate that here to bypass the “Comments to the Author” section, enter your conflict of interest statement in the “Confidential to Editor” section, and submit your "Accept" recommendation.

Reviewer #2: (No Response)

Reviewer #3: All comments have been addressed

2. Is the manuscript technically sound, and do the data support the conclusions?

Reviewer #2: Yes

Reviewer #3: Yes

3. Has the statistical analysis been performed appropriately and rigorously? 

Reviewer #2: Yes

Reviewer #3: Yes

4. Have the authors made all data underlying the findings in their manuscript fully available?

Reviewer #2: Yes

Reviewer #3: Yes

5. Is the manuscript presented in an intelligible fashion and written in standard English?

Reviewer #2: Yes

Reviewer #3: Yes

6. Review Comments to the Author

Reviewer #2: General Comments:

Please comment on how removing all warnings not to use cephalosporins in the setting of a penicillin allergy would affect your modeling as had been done at Kaiser Permanente. Please note that removing the warning still did not significantly reduce the increase morbidity associated with an unconfirmed penicillin allergy. It did result in the increased use of cephalosporins with no increased risk of antibiotic-associated hypersensitivity reactions.

https://pubmed.ncbi.nlm.nih.gov/33914051/

Specific Comments:

Lines 19 to 20 and Lines 47 to 49: Please make these lines agree with each other. Consider using the following in both places: “Less than five percent of patients who report a penicillin allergy will have a currently active clinically-significant IgE- or T-cell-mediated hypersensitivity

when appropriately tested.”

Lines 32 to 34: Please consider changing to “…person. Our modeling demonstrated that a test then treat strategy increased the probability of appropriate antibiotic use from 7,843/10,000 to 10,000/10,000 simulations.

Lines 38 to 41: Please consider changing to “Penicillin allergy is the most commonly reported allergy in pregnant women. The reported rate of unconfirmed penicillin allergy of about 8.0% in pregnant women is lower than commonly reported rates in outpatients using healthcare or other hospitalized populations.

Reviewer #3: (No Response)

7. PLOS authors have the option to publish the peer review history of their article (what does this mean?). If published, this will include your full peer review and any attached files.

Reviewer #2: **Yes: **Eric Macy

Reviewer #3: No

---

## [Author Response · Author response to Decision Letter 1]

14 Dec 2022

Reviewer #2: General Comments:

Please comment on how removing all warnings not to use cephalosporins in the setting of a penicillin allergy would affect your modeling as had been done at Kaiser Permanente. Please note that removing the warning still did not significantly reduce the increase morbidity associated with an unconfirmed penicillin allergy. It did result in the increased use of cephalosporins with no increased risk of antibiotic-associated hypersensitivity reactions.

https://pubmed.ncbi.nlm.nih.gov/33914051/

Thank you for bringing this publication to our attention. While we find it to be interesting, we do feel that we adequately accounted for the issue of over-vigilance regarding cephalosporin cross-reactivity in our model and in the text of the manuscript. In accordance with ACOG recommendations, at baseline our decision tree uses only anaphylaxis to penicillins or documented allergy to cephalosporins as a reason not to use cephalosporins in the non-intervention arm of the analysis. Given that, implementing the above warning would not change the outcome of our cost analysis. Additionally, our health system has no warnings in place regarding use of cephalosporins in penicillin allergic patients, so we do not feel the intervention in the above manuscript is broadly applicable regarding GBS prophylaxis. Finally, the removal of such warnings did not affect morbidity in the referenced manuscript and is also unlikely to affect cost in the framework of our decision analysis. For these reasons and in the interest of manuscript lenth we have declined to add the above discussion to this manuscript.

Specific Comments:

Lines 19 to 20 and Lines 47 to 49: Please make these lines agree with each other. Consider using the following in both places: “Less than five percent of patients who report a penicillin allergy will have a currently active clinically-significant IgE- or T-cell-mediated hypersensitivity

when appropriately tested.”

Thank you for that suggestion. Lines 19-20 have been changed to the language above, which matches that in lines 47-49.

Lines 32 to 34: Please consider changing to “…person. Our modeling demonstrated that a test then treat strategy increased the probability of appropriate antibiotic use from 7,843/10,000 to 10,000/10,000 simulations.

 This language has been changes as suggested.

Lines 38 to 41: Please consider changing to “Penicillin allergy is the most commonly reported allergy in pregnant women. The reported rate of unconfirmed penicillin allergy of about 8.0% in pregnant women is lower than commonly reported rates in outpatients using healthcare or other hospitalized populations.

 This sentence has been changed to use the above language.

---

## [Editor Report · Decision Letter 2]

21 Dec 2022

EVALUATING THE COST-EFFECTIVENESS OF TESTING PREGNANT WOMEN FOR PENICILLIN ALLERGY

PONE-D-22-17964R2

Dear Dr. Theiler,

We’re pleased to inform you that your manuscript has been judged scientifically suitable for publication and will be formally accepted for publication once it meets all outstanding technical requirements.

Kind regards,

Tefera Chane Mekonnen, Master in Public Health(MPH)

Academic Editor

PLOS ONE
---

## [Editor Report · Acceptance letter]

3 Jan 2023

PONE-D-22-17964R2 

Evaluating the cost-effectiveness of testing pregnant women for penicillin allergy 

Dear Dr. Theiler:

I'm pleased to inform you that your manuscript has been deemed suitable for publication in PLOS ONE. Congratulations! Your manuscript is now with our production department. 

Kind regards, 

on behalf of

Dr. Tefera Chane Mekonnen 

Academic Editor

PLOS ONE